# Quality of life of patients with solid malignancies at 3 months after unplanned admission in the intensive care unit: A prospective case-control study

**Anne-Claire Toffart**[1,2☯]*, **Wassila M'Sallaoui**[2☯], **Sophie Jerusalem**[3☯], **Alexandre Godon**[3‡], **Francois Bettega**[4‡], **Gael Roth**[5‡], **Julien Pavillet**[6‡], **Edouard Girard**[7‡], **Louis Marie Galerneau**[8‡], **Juliette Piot**[3‡], **Carole Schwebel**[8‡], **Jean Francois Payen**[3☯]

**1** Institute for Advanced Biosciences INSERM U1209 CNRS UMR5309, Grenoble Alpes University, Grenoble, France, **2** Department of Pneumology and Physiology, Grenoble Alpes University, Grenoble, France, **3** Department of Anesthesia and Intensive Care, Grenoble Alpes University, Grenoble, France, **4** HP2 Laboratory, INSERM U1300, Grenoble Alpes University, Grenoble, France, **5** Institute for Advanced Biosciences, CNRS UMR 5309/INSERM U1209, Grenoble Alpes University, Grenoble, France, **6** Department of Oncology, Grenoble Alpes University, Grenoble, France, **7** Department of Digestive and Emergency Surgery, Grenoble Alpes University, TIMC laboratory, CNRS, CHU Grenoble Alpes, Grenoble, France, **8** Department of Medical ICU, Grenoble Alpes University, Grenoble, France

☯ These authors contributed equally to this work.

‡ AG, FB, GR, JP, EG, LMG, JP and CS also contributed equally to this work.

* AToffart@chu-grenoble.fr

## Abstract

### Background

Although short- and long-term survival in critically ill patients with cancer has been described, data on their quality of life (QoL) after an intensive care unit (ICU) stay are scarce. This study aimed to determine the impact of an ICU stay on QoL assessed at 3 months in patients with solid malignancies.

### Methods

A prospective case-control study was conducted in three French ICUs between February 2020 and February 2021. Adult patients with lung, colorectal, or head and neck cancer who were admitted in the ICU were matched in a 1:2 ratio with patients who were not admitted in the ICU regarding their type of cancer, curative or palliative anticancer treatment, and treatment line. The primary endpoint was the QoL assessed at 3 months from inclusion using the mental and physical components of the Short Form 36 (SF-36) Health Survey. The use of anticancer therapies at 3 months was also evaluated.

### Results

In total, 23 surviving ICU cancer patients were matched with 46 non-ICU cancer patients. Four patients in the ICU group did not respond to the questionnaire. The mental component score of the SF-36 was higher in ICU patients than in non-ICU patients: median of 54

**Data Availability Statement:** Data cannot be shared publicly because they contain potentially identifying or sensitive patient information. Data

are available from the Grenoble Alpes University Hospital Institutional Data Access (contact via DRCI@chu-grenoble.fr) for researchers who meet the criteria for access to confidential data.

**Funding:** The author(s) received no specific funding for this work.

**Competing interests:** The authors have declared that no competing interests exist.

**Abbreviations:** ADL, activities of daily living; CI, confidence interval; ECOG, Eastern Cooperative Oncology Group; HADS, Hospital Anxiety and Depression Scale; ICU, intensive care unit; IQR, interquartile range; MCS, Mental Component Summary; OR, odds ratio; PCS, Physical Component Summary; PS, performance status; QoL, quality of life; SAPSII, Simplified Acute Physiology Score II; SF-36, Short Form-36; SOFA, Sequential Organ Failure Assessment; TLT, time-limited treatment.

(interquartile range: 42–57) *vs.* 47 (37–52), respectively ($p = 0.01$). The physical component score of the SF-36 did not differ between groups: 35 (31–47) *vs.* 42 (34–47) ($p = 0.24$). In multivariate analysis, no association was found between patient QoL and an ICU stay. A good performance status and a non-metastatic cancer at baseline were independently associated with a higher physical component score. The use of anticancer therapies at 3 months was comparable between the two groups.

## Conclusion

In patients with solid malignancies, an ICU stay had no negative impact on QoL at 3 months after discharge when compared with matched non-ICU patients.

## Introduction

Over the last decades, improved management of cancer patients [1–3] has led to an increase in their likelihood of unplanned intensive care unit (ICU) admission. Cancer patients occupy 15–20% of ICU beds [4, 5]. Reasons for ICU admission include cancer-related complications, anticancer treatment-related adverse effects, or organ failures unrelated to cancer.

Once admitted to the ICU, cancer patients were shown to have similar illness severity and outcomes as compared to non-cancer patients [6]. Previous studies found ICU and hospital mortality rates of 20–50% and 30–60%, respectively, in cancer patients [7–13]. Risk factors for mortality included poor Eastern Cooperative Oncology Group (ECOG) performance status (PS) prior to the admission, recurrent or progressive malignancy, and high Sequential Organ Failure Assessment (SOFA) score on admission [6]. New strategies have thus been developed for cancer patients who might need to be admitted to the ICU, such as early admission, non-invasive diagnostic and therapeutic options, and time-limited treatment (TLT) trials [14–19].

Short- and long-term quality of life (QoL) of patients was found to be impaired after an ICU stay compared to a matched non-ICU population [20–24]. The most pronounced reductions in QoL were observed after severe acute respiratory distress syndrome, prolonged mechanical ventilation, and severe sepsis [20]. However, data concerning the QoL of cancer patients after their discharge from the ICU are scarce. In a cohort of 483 critically ill patients with cancer, QoL was poorer at 3 and 12 months after ICU discharge [25]. Yet, the specific role of cancer in long-term QoL remains unclear. The present study aimed to determine the impact of an ICU stay on the 3-month QoL in patients with solid malignancies as compared to matched non-ICU patients.

## Materials and methods

This prospective case-control study was conducted between February 2020 and February 2021 within three adult ICUs of a French university hospital: a surgical ICU (19 beds), a cardio-surgical ICU (20 beds), and a medical ICU (28 beds). The study was approved by the Clinical Research Ethics Committee of the Rhône-Alpes-Auvergne region (IRB 5891) on February 20th, 2020 and, given its observational nature, the requirement for written informed consent from patients or relatives was waived. Each patient or his/her relatives received oral and written information about the research and could refuse to participate at any time in accordance with French law [26]. The study was registered on ClinicalTrials.gov (NCT04310033).

## Participants

Adult patients were included if they had lung, colorectal, or head and neck cancer (the three most common solid tumors encountered at the ICU) [27] and required acute admission to the ICU for a foreseeable stay of more than 24h (ICU group). Patients were identified upon ICU discharge and included in the study 5 to 14 days thereafter.

The decision of ICU admission was based on patient-specific criteria, taking into account known prognostic factors. There were no standardized pre-existing criteria.

Patients were not included if they had a solid malignancy in remission for more than 2 years, if they did not have sufficient French language skills for telephone interviews, or if they or a relative opposed their participation in the study.

This ICU group was matched in a 1:2 ratio with a group of patients who were not admitted in the ICU (non-ICU group) regarding their type of cancer, curative or palliative anticancer treatment, and treatment line. For this purpose, we used an algorithm for prospective individual matching according to Charpentier et al. [28] (Fig 1). Non-ICU patients were prospectively identified from the oncological consultation to form an unmatched pool of non-ICU patients of each cancer type (lung, colorectal, or head and neck). Each included ICU patient was simultaneously matched with two non-ICU patients. If such non-ICU patients were not available, the ICU patient was put in the pool of ICU patients to be potentially matched with two non-ICU patients later (Fig 1). If an ICU patient died prior to the 3-month assessment, the two matched non-ICU patients returned to the pool of non-ICU patients; if a non-ICU patient

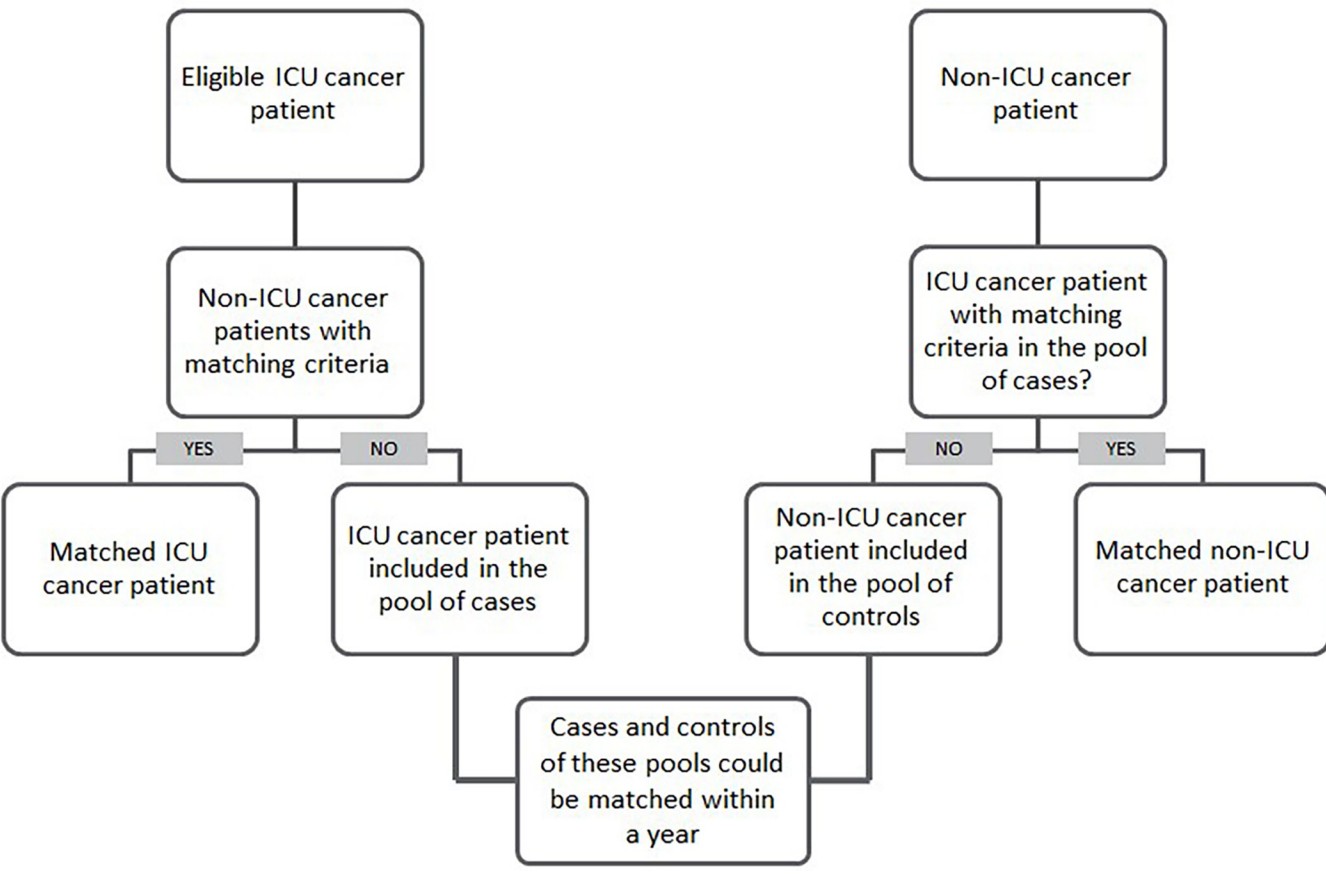

**Fig 1. Method of prospective individual matching.**

died, he/she was replaced by another matched non-ICU patient. If an ICU-patient did not answer at 3 months, he/she was excluded from the analysis, along with the two matched non-ICU patients.

### Data collection

Data collected included patient characteristics prior to ICU admission, solid malignancy characteristics, and ICU stay details recorded from admission day (Day 1) to discharge. Severity scores were recorded in the ICU: Simplified Acute Physiology Score II (SAPSII) [29], and Sepsis-related Organ Failure Assessment (SOFA) [30]. Patients' status was assessed at 3 months during a telephone interview with the patient or a relative, which was conducted by a trained physician (SJ, WM).

### Endpoints

The primary outcome was the QoL at 3 months after discharge from the ICU (or equivalent) as assessed by the 36-Item Short-Form Health Survey (SF-36), which the patients answered during a telephone interview.

The eight SF-36 domains incorporate two dimensions: the Physical Component Summary (PCS) and the Mental Component Summary (MCS). The PCS is composed of four scales assessing physical function, role of limitations due to physical problems, bodily pain, and general health. The MCS is composed of four scales assessing social functioning, role of limitations due to emotional problems, mental health, and vitality. The two components range from 0 to 100, with higher scores indicating better perceived health [31–34]. The PS to estimate the patient's ability to perform activities of daily living (ADL) was determined using the ECOG scale, ranging from 0 (fully functional) to 4 (bedridden). Self-sufficiency was assessed using the Katz Index of Independence in ADL [35]. An ADL score of 6 indicated full independence, and a score of 2 or less high dependence on daily activities. Anxiety and depression were evaluated using the Hospital Anxiety and Depression Scale (HADS) including a score for anxiety (HADS-A) and another for depression (HADS-D), each ranging from 0 to 21 (scores of less than 7 indicate non-cases). The continuation or discontinuation of anticancer treatment at 3 months was also determined.

### Statistical analysis

Data were expressed as number and percentage, or median and interquartile range (25th–75th percentile). The primary outcome analysis included all patients who answered the QoL questionnaire at 3 months. Comparisons between the two groups (ICU group *vs*. non-ICU group) were performed using a Chi-squared test, Fisher's exact test, or non-parametric Mann-Whitney U test, where appropriate. A multivariate analysis was performed using a generalized linear model with a gamma distribution of factors not used for matching, i.e. age and gender, and clinically relevant variables, i.e. PS and metastatic cancer status. The statistical significance level was set at $p<0.05$ and statistical analyses were conducted using R software version 4.0.3.

Based on the literature, we hypothesized that the mean SF-36 score at 3 months would be 35/100 in ICU cancer patients *vs*. 45/100 in non-ICU cancer patients [21, 25, 36, 37]. Assuming a statistical power of 90%, an alpha risk of 5%, and a rate of lost to follow-up patients of 20%, 20 ICU patients and 40 matched non-ICU patients would be enough to reliably assess the impact of an ICU stay on QoL at 3 months.

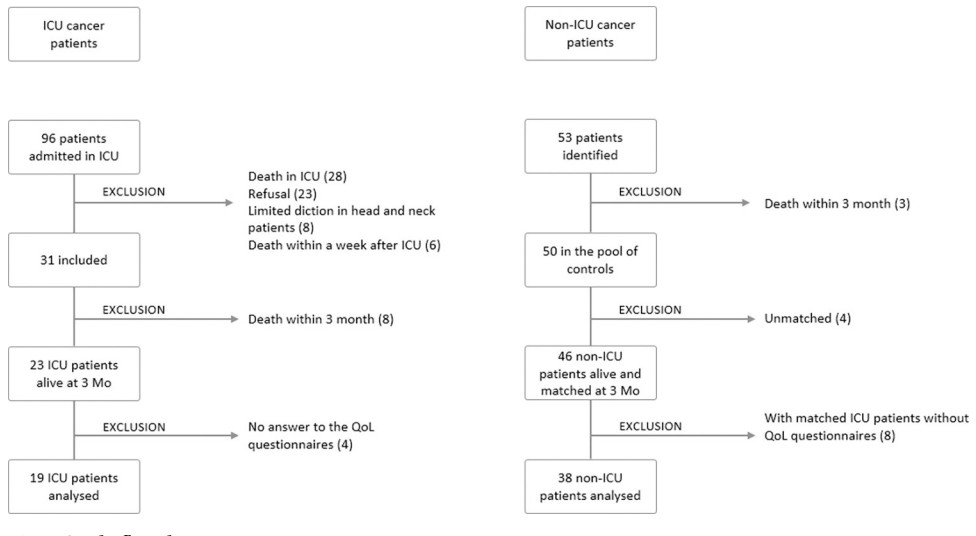

**Fig 2. Study flowchart.**

## Results

Of 96 cancer patients (lung, colorectal, head and neck cancers) discharged from the ICU, 31 were included and 23 were still alive 3 months after their discharge from the ICU (ICU group). They were matched with 46 non-ICU patients with solid malignancies (non-ICU group) (Fig 2).

Most ICU and non-ICU patients had lung cancer (n = 39/69 patients) and metastatic cancer (34/69 patients). There were 30/69 patients who received curative anticancer treatment. The two patient groups were comparable regarding several variables, but ICU patients had lower PS at baseline (Table 1).

The ICU stay characteristics of the 23 ICU patients are shown in Table 2. Most of them were admitted for cancer-related complications and had acute respiratory failure on presentation.

### Quality of life at 3 months

In total, 19 ICU patients and 46 non-ICU patients answered the QoL questionnaire at 3 months after ICU discharge (or equivalent). The median MCS value was higher in ICU patients than in non-ICU patients: 54 [42–57] *vs*. 47 [37–52], respectively (*p* = 0.01). The median PCS value did not significantly differ between groups (Table 3).

In multivariate analysis, non-metastatic cancer and good PS at baseline were two factors associated with an increase in PCS scores (Table 4).

### General condition and anticancer treatment

At 3 months, the number of patients with a good PS (0 to 1) was significantly lower in ICU than in non-ICU patients: 18 *vs*. 45 patients, respectively (*p* = 0.01) (Table 3). However, the HADS-A was significantly better in ICU than in non-ICU patients: 5 [4–6] *vs*. 8 [5–10], respectively (*p* = 0.01). The HADS-D and ADL scores were comparable between the two groups (Table 3).

At 3 months, for 16/23 (70%) ICU patients and 40/46 (97%) non-ICU patients, cancer care was not modified compared to baseline (*p* = 0.01) (Table 3). Among ICU patients, two were in exclusive palliative care at 3 months (none in the control group).

**Table 1. Baseline characteristics of patients.**

| | ICU patients, n = 23 | Non-ICU patients, n = 46 | *p*-value |
|---|---|---|---|
| **Demographic data** | | | |
| Age (in years) | 70 [60;75] | 66.5 [60;75] | 0.91 |
| Male gender | 18 | 27 | 0.18 |
| Weight at inclusion (kg) | 62 [55;69] | 68 [58;79] | 0.26 |
| Charlson comorbidity index | 1 [0;3] | 2 [0;3] | 0.32 |
| Performance status | | | |
| • 0–1 | 18 | 46 | <0.01 |
| • 2–3 | 5 | | <0.01 |
| **History of cancer** | | | |
| Primitive cancer | | | * |
| • Lung | 13 | 26 | |
| • Colorectal | 6 | 12 | |
| • Head and neck | 4 | 8 | |
| Non-metastatic disease | 12 | 23 | 1 |
| Previous anticancer treatments | | | |
| • Chemotherapy | 15 | 30 | 1 |
| • Immunotherapy | 6 | 13 | 1 |
| • Curative radiotherapy | 5 | 12 | 0.92 |
| • Curative surgery | 10 | 20 | 1 |
| • Targeted therapy | 3 | 5 | 1 |
| **Anticancer treatment on admission** | | | |
| Treatment goal | | | * |
| • Palliative | 13 | 26 | |
| • Curative | 10 | 20 | |
| Treatment line | | | * |
| • Only surgery or radiotherapy | 6 | 12 | |
| • Line 1 of systemic treatment | 14 | 28 | |
| • Line 2–3 of systemic treatment | 3 | 6 | |
| Cancer status | | | ** |
| • Treatment not reassessed | 10 | 19 | |
| • Controlled without treatments | 7 | 11 | |
| • Controlled with treatments | 5 | 16 | |
| • Progression | 1 | 0 | |

* Not performed because of matching variables.

** Not meeting test assumption.

ICU: intensive care unit; non-ICU: non-intensive care unit.

Data are expressed as median (25$^{th}$–75$^{th}$ percentile) or number.

## Discussion

This prospective case-control study compared the QoL of cancer patients after being admitted (n = 19) or not admitted (n = 38) to the ICU. At 3 months, the MCS of the SF-36 was significantly higher in ICU patients. After adjustment, there was no significant association between an ICU admission and patient QoL. These findings indicate that an ICU stay had no major impact on QoL at 3 months in patients with solid malignancies.

The MCS values at 3 months were higher in ICU patients than in non-ICU patients. Of note, the MCS was higher than the PCS in the ICU group, as found elsewhere [25, 37]. These

**Table 2. ICU stay characteristics of the 23 ICU patients.**

| | |
|---|---|
| **Reason for admission** | |
| • Cancer-related complications | 6 |
| • Cancer treatment adverse effects | 9 |
| • Not related to cancer | 8 |
| **Organ failure on admission** [*] | |
| • Respiratory | 13 |
| • Cardiovascular | 9 |
| • Coma | 2 |
| • Sepsis | 3 |
| **Severity scores** | |
| • SAPS II | 43 [33;51] |
| • Worst SOFA | 4 [3;6] |
| **Number of organ failures** | 1 [1;2] |
| **Organ support** | |
| • Vasopressors | 9 (39) |
| • Invasive ventilation | 9 (39) |
| • Renal replacement therapy | 0 |
| **Length of ICU stay (days)** | 4 [1.5;7] |
| **Length of hospital stay (days)** | 12 [7;17] |

[*] Not mutually exclusive. ICU: intensive care unit; SAPSII: Simplified Acute Physiology Score II; SOFA: Sequential Organ Failure Assessment.

Data are expressed as median (25th–75th percentile) or number.

findings can be explained by the psychological care these patients received after an ICU stay, which might help them to progressively adapt and accept their physical limitations. Oeyen et al. showed that the 3-month QoL of solid cancer patients declined significantly and did not return to baseline at 12 months after ICU discharge [25]. However, the evolution of cancer

**Table 3. Patient QoL and anticancer treatments at 3 months.**

| | ICU patients, n = 23 | Non-ICU patients, n = 46 | *p*-value |
|---|---|---|---|
| **Demographic data** | | | |
| Change in weight (%) | 1.3 [-4;3.9] | 0 [-1.6;2.7] | 0.9 |
| Change in performance status | 0 [-1; 0.5] | 0 [0;0] | 0.52 |
| **Quality of life at 3 months** | On 19 patients | On 38 patients | |
| Mental SF-36 | 54 [42;57] | 47 [37;52] | 0.01 |
| Physical SF-36 | 35 [31;47] | 42 [34;47] | 0.24 |
| HADS-A | 5 [4;6] | 8 [5;10] | 0.01 |
| HADS-D | 3 [2;6] | 5 [2;8] | 0.29 |
| ADL | 6 [6;6] | 6 [6;6] | 0.6 |
| **Anticancer treatment at 3 months** | | | |
| • Continued | 16 | 40 | 0.1 |
| • Additional treatment line | 5 | 6 | 0.5 |
| • Exclusive palliative care | 2 | 0 | 0.1 |

SF-36; Short-Form 36; HADS: Hospital Anxiety and Depression Scale; ADL: activities of daily living; ICU: intensive care unit.

Data are expressed as median (25th-75th percentile) or number.

**Table 4. Multivariate analysis of variables associated with SF-36 at 3 months.**

| | Mental SF-36 | | | Physical SF-36 | | |
|---|---|---|---|---|---|---|
| | Coef. | 95%CI | *p*-value | Coef. | 95%CI | *p*-value |
| **Case-control** | | | | | | |
| • controls | 1 | — | | 1 | — | |
| • cases | 1.11 | 0.97–1.27 | 0.1 | 1.01 | 0.86–1.19 | >0.9 |
| **Age** | 1 | 0.99–1.01 | 0.9 | 1 | 0.99–1.00 | 0.3 |
| **Gender** | | | | | | |
| • F | 1 | — | | 1 | — | |
| • H | 1.05 | 0.93–1.18 | 0.5 | 0.89 | 0.77–1.03 | 0.13 |
| **Performance status at admission** | | | | | | |
| • 2–3 | 1 | — | | 1 | — | |
| • 0–1 | 0.99 | 0.79–1.23 | >0.9 | 1.46 | 1.11–1.90 | 0.01 |
| **Non-metastatic** | 0.95 | 0.85–1.07 | 0.4 | 1.21 | 1.05–1.40 | 0.01 |

Coef.: regression coefficient; CI: confidence Interval

over time might have played a role in this gradual decline in QoL. To address this issue, we chose to document the short-term impact of an ICU stay in patients with solid malignancies in a prospective case-control study with matched non-ICU cancer patients, and we found no deleterious effect of an ICU stay on QoL at 3 months.

The present study was conducted in a highly selected population and included patients still alive at 3 months after ICU discharge. Of note were the good baseline PS of our patients and their low SOFA scores during the ICU stay (Tables 1 and 2). These two factors have been acknowledged to be independently associated with high survival rates in critically ill patients with cancer [6, 7, 9–12, 27, 38, 39]. In a retrospective study conducted in cancer patients with suspected infection, a poor PS was an additional factor for increased mortality in patients with SOFA scores of 6 or less [40]. In addition, non-recurrent/-progressive cancer disease was associated with a better survival rate at 6 months post-ICU [40]. Accordingly, the present study showed that a good PS at baseline and a non-metastatic cancer status might both have contributed to the absence of negative impact of an ICU stay on QoL. These two factors could be taken into consideration every time a possible ICU admission of a solid cancer patient is discussed.

Once admitted in the ICU, a patient with solid malignancy may be exposed to an additional risk of poor outcome due to cancer-related factors, such as anticancer treatments prior to admission or immune system disorders. The incidence of severe infections acquired during an ICU stay was found higher in this population compared to non-cancer patients [41]. Despite mild-to-moderate SAPSII scores on admission (Table 2) and a mortality rate of 26% at 3 months, our patients did not develop multiple or severe organ failures, as reflected by their low SOFA scores during the ICU stay. We have recently shown that a SOFA score of 6 or more on Day 4 was significantly associated with mortality on Day 90 in octogenarians after their acute admission in the ICU [42]. Scoring SOFA within a few days of ICU admission may help physicians to objectively measure the response to therapy as part of the TLT concept [42]. In critically ill patients with solid tumors, a TLT duration of 1 to 4 days after ICU admission using SOFA scores to assess the patient's condition was found to be sufficient to achieve optimal survival benefit [18]. Consequently, the discussion around the clinical appropriateness of continuing treatments in critically ill cancer patients could be based in part on the short-term response to therapy.

This study has several limitations. First, the study was conducted in three sites of one university hospital, with a small number of patients. Although the original study design with prospective recruitment of matched patients avoided selection bias or missing data, our findings cannot be extrapolated to other sites due to the limited size of the study population. Second, QoL was not assessed at baseline, i.e. prior to ICU admission or equivalent, to measure a possible change in the SF-36 at 3 months in each group of patients. It should be noted, however, that the MCS and PCS of our patients were similar to those found elsewhere at 3 months [25]. Third, our ICU patients had low SOFA scores probably because of the predefined selection criteria to conduct the study and their good PS at baseline. Whether similar results could be obtained in more severe ICU patients warrants further investigation.

In conclusion, patients with solid malignancies did not display impaired QoL at 3 months after ICU discharge when compared to matched patients not admitted in the ICU. A good PS and non-metastatic disease prior to admission could have contributed to these findings.

## Author Contributions

**Conceptualization:** Anne-Claire Toffart, Wassila M'Sallaoui, Sophie Jerusalem, Alexandre Godon, Jean Francois Payen.

**Data curation:** Wassila M'Sallaoui, Sophie Jerusalem.

**Funding acquisition:** Wassila M'Sallaoui, Sophie Jerusalem, Gael Roth, Julien Pavillet, Edouard Girard, Louis Marie Galerneau, Juliette Piot, Carole Schwebel.

**Methodology:** Anne-Claire Toffart, Francois Bettega, Gael Roth, Julien Pavillet, Edouard Girard, Louis Marie Galerneau, Juliette Piot, Carole Schwebel.

**Project administration:** Anne-Claire Toffart.

**Supervision:** Anne-Claire Toffart, Alexandre Godon, Jean Francois Payen.

**Validation:** Anne-Claire Toffart, Jean Francois Payen.

**Writing – original draft:** Anne-Claire Toffart, Wassila M'Sallaoui, Sophie Jerusalem.

**Writing – review & editing:** Anne-Claire Toffart, Alexandre Godon, Gael Roth, Julien Pavillet, Edouard Girard, Louis Marie Galerneau, Juliette Piot, Carole Schwebel, Jean Francois Payen.

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
