## [Decision Letter · Decision Letter 0]

18 Oct 2022

PONE-D-22-19984Quality of life of patients with solid malignancies at 3 months after unplanned admission in the intensive care unit: A prospective case-control studyPLOS ONE

Dear Dr. Toffart,

Thank you for submitting your manuscript to PLOS ONE. After careful consideration, we feel that it has merit but does not fully meet PLOS ONE’s publication criteria as it currently stands. Therefore, we invite you to submit a revised version of the manuscript that addresses the points raised during the review process.

We look forward to receiving your revised manuscript.

Kind regards,

Raphael Mendonça Guimaraes, PhD

Academic Editor

PLOS ONE

Journal Requirements:

2. PLOS requires an ORCID iD for the corresponding author in Editorial Manager on papers submitted after December 6th, 2016. Please ensure that you have an ORCID iD and that it is validated in Editorial Manager. To do this, go to ‘Update my Information’ (in the upper left-hand corner of the main menu), and click on the Fetch/Validate link next to the ORCID field. This will take you to the ORCID site and allow you to create a new iD or authenticate a pre-existing iD in Editorial Manager. Please see the following video for instructions on linking an ORCID iD to your Editorial Manager account: https://www.youtube.com/watch?v=_xcclfuvtxQ.

3. Please amend the manuscript submission data (via Edit Submission) to include author “W M’Sallaoui”

4. Please amend your authorship list in your manuscript file to include author “Juliette Piot, Wassila Marnas”

Reviewers' comments:

Reviewer's Responses to Questions

**Comments to the Author**

1. Is the manuscript technically sound, and do the data support the conclusions?

Reviewer #1: Yes

Reviewer #2: Yes

2. Has the statistical analysis been performed appropriately and rigorously? 

Reviewer #1: Yes

Reviewer #2: Yes

3. Have the authors made all data underlying the findings in their manuscript fully available?

Reviewer #1: Yes

Reviewer #2: No

4. Is the manuscript presented in an intelligible fashion and written in standard English?

Reviewer #1: No

Reviewer #2: Yes

5. Review Comments to the Author

Reviewer #1: Thank you for the opportunity to review the paper “Quality of life of patients with solid malignancies at 3 months after unplanned admission in the intensive care unit: a prospective case-control study”. I have included some comments that I hope may be of use to the authors.

Methods:

I wonder if there are pre-existing criteria for admission to the ICU at the authors’ organization, e.g. based on performance status; co-morbidities; use of TLT etc. If so, it would be helpful to describe these in this section.

How were patients identified? At the time of admission to the ICU, or after discharge? How soon afterwards were they approached about the study?

Who completed the SF-36, patients or their family members? If the latter, this should be described in the paper.

Can the authors speak to their inclusion criteria of lung, colorectal and head and neck cancers only?

Several other studies have shown that social supports influenced QOL after an ICU stay. Did the authors consider including this as a potential factor for this study?

Results:

How many patients with cancer were admitted to the ICU during the study period (i.e. how many died in the ICU)?

Only a small number of patients had colorectal or head and neck cancers. Were these less likely to be admitted to the ICU in the first place?

Consider including percentages in the tables for clarity.

I think there is an error in Table 1, where non-metastatic disease adds up to 35. I think it should be 34 (35/69 had metastatic disease according to the text)?

Include an explanation for the IGS2 and SOFA scores for readers who may not be familiar with these.

There is a typo in Table 3. It says ‘change in eight’ instead of ‘change in weight’.

Discussion:

‘These findings can be explained by the psychological care these patients received after a stay in the ICU’…can the authors elaborate on exactly what support these patients received?

‘The incidence of severe infections acquired during the ICU stay was found [to be] higher in this population compared to non-cancer patients’. This sentence is unclear. Does it refer to the current study, in which case this information might be better presented in the results section? Alternatively, does it refer to patients with cancer admitted to the ICU in general, in which case it needs a reference?

General comments

The small sample size and heterogeneity in terms of cancer diagnoses and stage of illness makes the findings difficult to interpret or generalize.

The absence of baseline QOL scores is also a challenge, in that the ICU group may have had higher QOL scores to begin with.

Additional demographic factors such as marital status, social supports, symptom burden might be useful in future studies exploring QOL.

There are quite a few grammatical errors throughout the paper. If considered for publication, I would recommend a native English speaker familiar with medical writing review it.

Reviewer #2: I congratulate the authors for the well-designed and written manuscript that presents important contributions to the care of cancer patients admitted to the ICU. The limitations were well described and the conclusion adequately responds to the objectives.

6. PLOS authors have the option to publish the peer review history of their article (what does this mean?). If published, this will include your full peer review and any attached files.

Reviewer #1: No

Reviewer #2: No

---

## [Author Response · Author response to Decision Letter 0]

28 Nov 2022

REPLY TO THE EDITOR:

28th November 2022

Dear Editor,

We would like to express our full appreciation for giving us the opportunity to revise our manuscript Ref. PONE-D-22-19984. We believe that the reviewers made insightful suggestions that greatly helped to improve the manuscript. In red you will find the changes taking into account remarks of reviewers and highlighted in yellow the corrected English

We expect that this manuscript can be now suitable for publication in PLOS One. We remain at your disposal to clarify any pending point.

Yours sincerely,

Wassila Marnas, MD

Anne-Claire Toffart, MD, PhD

 

Answers to Journal Requirements

R : Done

2. PLOS requires an ORCID iD for the corresponding author in Editorial Manager on papers submitted after December 6th, 2016. Please ensure that you have an ORCID iD and that it is validated in Editorial Manager. To do this, go to ‘Update my Information’ (in the upper left-hand corner of the main menu), and click on the Fetch/Validate link next to the ORCID field. This will take you to the ORCID site and allow you to create a new iD or authenticate a pre-existing iD in Editorial Manager. Please see the following video for instructions on linking an ORCID iD to your Editorial Manager account: https://www.youtube.com/watch?v=_xcclfuvtxQ.

R : https://orcid.org/0000-0003-4377-0227

3. Please amend the manuscript submission data (via Edit Submission) to include author “W M’Sallaoui”

R : W M’Sallaoui is Wassila Marnas. We modified in the Edit Submission Marnas in M’Sallaoui

4. Please amend your authorship list in your manuscript file to include author “Juliette Piot, Wassila Marnas”

R : Wassila Marnas is W M’Sallaoui. Juliette Piot is still in our authorship list. We modified tis authorship list as requested by Plos One.

R : Data cannot be shared publicly because of contains potentially identifying or sensitive patient information. Data are available from the Grenoble Alpes Universityt Hospital Institutional Data Access (contact via DRCI@chu-grenoble.fr) for researchers who meet the criteria for access to confidential data.

R : Done. We have not modified our reference list.

Answers to Reviewers’ Comments: 

4. Is the manuscript presented in an intelligible fashion and written in standard English?

Reviewer #1: No

Reviewer #2: Yes

R : Done, we reviewed English.

Reviewer #1: 

Methods:

I wonder if there are pre-existing criteria for admission to the ICU at the authors’ organization, e.g. based on performance status; co-morbidities; use of TLT etc. If so, it would be helpful to describe these in this section.

R : The decision to admit to ICU is based on the criteria of each patient, taking into account known prognostic factors. There are no standardized pre-existing criteria. 

We added these sentences in the manuscript (page 6, §2)

How were patients identified? At the time of admission to the ICU, or after discharge? 

R : Patients were identified at ICU discharge.

We added “Patients were identified at ICU discharge, and included between 5 and 14 days after ICU discharge.” (page 6, §2)

How soon afterwards were they approached about the study?

R : Patients were included between 5 and 14 days after ICU discharge.

We added : “Patients were identified at ICU discharge, and included between 5 and 14 days after ICU discharge.” (page 6, §2)

Who completed the SF-36, patients or their family members? If the latter, this should be described in the paper.

R : The SF-36 was completed by the patients during a telephone interview

We added : « which the patients answered during a telephone interview. » (page 8, §2) 

Can the authors speak to their inclusion criteria of lung, colorectal and head and neck cancers only?

R : Based on a previous publication “Gheerbrant H, Timsit JF, Terzi N, Ruckly S, Laramas M, Levra MG, et al. Factors associated with survival of patients with solid Cancer alive after intensive care unit discharge between 2005 and 2013. BMC Cancer. 5 janv 2021;21(1):9.”, we identified lung, colorectal and head and neck cancers as the most frequent cancers admitted in our ICUs. 

We added: “Adult patients were included if they had lung, colorectal, or head and neck cancer (the three most common solid tumors encountered at the ICU) [27] “(page 6, §2)

Several other studies have shown that social supports influenced QOL after an ICU stay. Did the authors consider including this as a potential factor for this study?

R : We agree. Unfortunately, we did not collect data on social conditions. As far as French law is concerned, the study would be more complicated to carry out.

Results:

How many patients with cancer were admitted to the ICU during the study period (i.e. how many died in the ICU)?

R : We did not collect this data because that was beyond the objectives of this study. In our institution, between 2005 and 2013, 30% of patients (108/361) with solid tumors died in ICU (Gheerbrant et al. BMC Cancer 2021).

Only a small number of patients had colorectal or head and neck cancers. Were these less likely to be admitted to the ICU in the first place?

R : We cannot answer to this question. The location of cancer is not a variable used in the decision of admission to the ICU. In the literature, cancer patients in the ICU were described with hematological malignancies, localized or metastatic solid tumor, not regarding the location of cancer.

Consider including percentages in the tables for clarity.

I think there is an error in Table 1, where non-metastatic disease adds up to 35. I think it should be 34 (35/69 had metastatic disease according to the text)?

R : We thank you for this comment. We modified the Table 1.

Include an explanation for the IGS2 and SOFA scores for readers who may not be familiar with these.

R : Thank you for your comment: IGS2 is a French abbreviation. We modified in SAPS II. SAPS II and SOFA scores were usually used to described severity of patients admitted in ICU

We added « Severity scores in ICU were recorded: Simplified Acute Physiology Score II (SAPSII) [29] and Sepsis-related Organ Failure Assessment (SOFA) scores [30] » (page 8, §1)

There is a typo in Table 3. It says ‘change in eight’ instead of ‘change in weight’.

R : You are right. Thank you for this comment. We modified the Table 3.

Discussion:

‘These findings can be explained by the psychological care these patients received after a stay in the ICU’…can the authors elaborate on exactly what support these patients received?

R : That is an hypothesis. We did not collect data on the psychological follow-up of patients.

‘The incidence of severe infections acquired during the ICU stay was found [to be] higher in this population compared to non-cancer patients’. This sentence is unclear. Does it refer to the current study, in which case this information might be better presented in the results section? Alternatively, does it refer to patients with cancer admitted to the ICU in general, in which case it needs a reference?

R : This sentence does not refer to the current study. Here is the reference that was included in the article : Vincent JL, Rello J, Marshall J, Silva E, Anzueto A, Martin CD, Moreno R, Lipman J, Gomersall C, Sakr Y, Reinhart K; EPIC II Group of Investigators. International study of the prevalence and outcomes of infection in intensive care units. JAMA. 2009 Dec 2;302(21):2323-9. doi: 10.1001/jama.2009.1754. PMID: 19952319.

General comments

The small sample size and heterogeneity in terms of cancer diagnoses and stage of illness makes the findings difficult to interpret or generalize.

R : We agree. This point has been added in the limits of the study.

The absence of baseline QOL scores is also a challenge, in that the ICU group may have had higher QOL scores to begin with.

R : We agree. As mentioned in the discussion, assessing QOL at baseline, i.e. prior to ICU admission or equivalent, was not possible. 

Additional demographic factors such as marital status, social supports, symptom burden might be useful in future studies exploring QOL.

R : We agree. According to the French law, collecting “marital status, social supports” are considered sensitive data. Collecting such data would have required a strong justification. This point has been mentioned in the Discussion. 

There are quite a few grammatical errors throughout the paper. If considered for publication, I would recommend a native English speaker familiar with medical writing review it.

R : Done

Reviewer #2: I congratulate the authors for the well-designed and written manuscript that presents important contributions to the care of cancer patients admitted to the ICU. The limitations were well described and the conclusion adequately responds to the objectives.

R : We thank you for this positive comment.

---

## [Editor Report · Decision Letter 1]

21 Dec 2022

Quality of life of patients with solid malignancies at 3 months after unplanned admission in the intensive care unit: A prospective case-control study

PONE-D-22-19984R1

Dear Dr. Toffart,

We’re pleased to inform you that your manuscript has been judged scientifically suitable for publication and will be formally accepted for publication once it meets all outstanding technical requirements.

Kind regards,

Raphael Mendonça Guimaraes, PhD

Academic Editor

PLOS ONE
---

## [Editor Report · Acceptance letter]

23 Dec 2022

PONE-D-22-19984R1 

Quality of life of patients with solid malignancies at 3 months after unplanned admission in the intensive care unit:
A prospective case-control study 

Dear Dr. Toffart:

I'm pleased to inform you that your manuscript has been deemed suitable for publication in PLOS ONE. Congratulations! Your manuscript is now with our production department. 

Kind regards, 

on behalf of

Dr. Raphael Mendonça Guimaraes 

Academic Editor

PLOS ONE